# Comparative Genomic Analysis and Physiological Properties of *Limosilactobacillus fermentum* SMFM2017-NK2 with Ability to Inflammatory Bowel Disease

**DOI:** 10.3390/microorganisms11030547

**Published:** 2023-02-21

**Authors:** Sumin Ann, Yukyung Choi, Yohan Yoon

**Affiliations:** 1Department of Food and Nutrition, Sookmyung Women’s University, Seoul 04310, Republic of Korea; 2Risk Analysis Research Center, Sookmyung Women’s University, Seoul 04310, Republic of Korea

**Keywords:** lactic acid bacteria, anti-inflammatory effect, IBD, comparative genomics

## Abstract

The objective of this study was to evaluate the anti-inflammatory effect of *Latilactobacillus sakei* SMFM2017-NK1 (LS1), *L. sakei* SMFM2017-NK3 (LS2), and *Limosilactobacillus fermentum* SMFM2017-NK2 (LF) on colitis using an animal model. DSS (dextran sulfate sodium salt) was orally injected into C57BL/6N mice to induce inflammation in the colon for seven days. Colitis mice models were treated with LS1, LS2, and LF, respectively, and *Lacticaseibacillus rhamnosus* GG (LGG) was used as a positive control. During oral administration of lactic acid bacteria, the weights of the mice were measured, and the disease activity index (DAI) score was determined by judging the degree of diarrhea and bloody stool. When comparing the differences between the minimum weight after DSS administration and the maximum weight after lactic acid bacteriaadministration were compared, the LF-treated group showed the highest weight gain at 8.91%. The DAI scores of the LF, LS2, and LGG groups were lower than that of the control group. After sacrifice, mRNA expression levels for proinflammatory cytokines (TNF-α, IL-1β, IL-6, and IFN-γ) and mediators (iNOS and COX-2) in the colon were measured. LF was selected as a superior strain for anti-inflammation in the colon. It was further analyzed to determine its biochemical characteristics, cytotoxicity, and thermal stability. Catalase and oxidase activities for LF were negative. In cytotoxicity and heat stability tests, the LF group had higher cell viability than the LGG group. The genome of LF was obtained, and 5682 CDS, 114 tRNA, 2 RNA, and 5 repeat regions were predicted. Especially, LF could be distinguished from the other three *L. fermentum* strains based on taxonomic profiling, specific orthologous genes of the strain, and genomic variants. The results of this study suggest that *L. fermentum* SMFM2017-NK2 is a novel strain with an anti-inflammatory effect on colitis.

## 1. Introduction

As the average life expectancy is extended, the number of patients with chronic diseases such as diabetes, hypertension, colorectal cancer, Crohn’s disease, and ulcerative colitis is also increased [1,2]. The incidence of inflammatory bowel disease (IBD) in Republic of Korea is also gradually increasing. Thus, it is important to prevent and treat IBD patients [3].

Functional foods rather than synthetic drugs are receiving attention for preventing or alleviating IBD. Among them, probiotics have been recognized as a good functional food because of their various functionalities such as intestinal immunity enhancement and body fat reduction [4]. In particular, a case study on the usefulness of probiotics has shown that *Lactobacillus* is associated with a reduce colorectal cancer risk [5]. Park et al. [6] have also shown the effectiveness of red pepper paste containing lactic acid bacteria isolated from kimchi on colitis in C57BL/6 mice. A decrease in ulcerative colitis disease activity index was achieved in 44.6% of VSL#3-treated patients versus 25.1% of patients given a placebo [7]. *Lactobacillus pentosus* Miny-148 isolated from human feces could inhibit the growth of HT-29 colon cancer cells and transmissible gastroenteritis virus [8]. According to these studies, lactic acid bacteria might be able to alleviate inflammation in the colon as probiotics.

To date, whether lactic acid bacteria could alleviate IBD remains unclear. Therefore, the objective of this study was to determine whether lactic acid bacteria could alleviate IBD and perform safety examinations for lactic acid bacteria followed by whole genome sequencing to determine their originality.

## 2. Materials and Methods

### 2.1. Preparation of Lactic Acid Bacteria

*Latilactobacillus sakei* SMFM2017-NK1 (LS1), *L. sakei* SMFM2017-NK3 (LS2), and *Limosillactobacillus fermentum* SMFM2017-NK2 (LF) are strains that showed antioxidant and inflammation effects [9]. *Lacticaseibacillus rhamnosus* GG (LGG; ATCC53103), known as having effects on diarrhea and anti-inflammation, was also used to compare its effect with lactic acid bacteria strains. A total of 100 μL of each strain in 20% glycerol stock was inoculated in 10 mL of sterilized MRS (de Man, Rogosa and Sharpe) broth (Becton Dickinson and Company, Sparks, MD, USA) at 37 °C for 24 h under anaerobic conditions, which was prepared by anaerogen™ (Oxoid, Hampshire, England, UK) in a tightly sealed container. After that, 100 μL of the culture medium was transferred into fresh 10 mL MRS broth and incubated at 37 °C for 24 h. Sub-cultures were centrifuged at 1912× *g* and 4 °C for 15 min and supernatants were discarded. Cell pellets were washed with phosphate buffer saline (PBS; pH 7.4; 0.2 g of KH_2_PO_4_, 1.5 g of Na_2_HPO_4_, 8.0 g of NaCl, 0.2 g of KCl in 1 L distilled water) twice, and then resuspended in PBS. For oral administration in mice, 9 Log CFU/mL of lactic acid bacteria were obtained.

### 2.2. Development of a Colitis Mouse Model and Administration of Lactic Acid Bacteria

Five-week-old C57BL/6N mice (body weight: 20 ± 2 g) (Raonbio, Yongin, Gyeonggi-do, Republic of Korea) were adapted under constantly controlled conditions (23 ± 3 °C, 55% ± 15% humidity). These mice were randomly divided into six groups (8 mice in each group): (1) normal (normal mice administered with PBS), (2) control (colitis-induced mice administered with PBS), (3) LS1 (colitis-induced mice administered with *L. sakei* SMFM2017-NK1), (4) LS2 (colitis-induced mice administered with *L. sakei* SMFM2017-NK3), (5) LF (colitis-induced mice administered with *L. fermentum* SMFM2017-NK2), and (6) LGG (colitis-induced mice administered with *L. rhamnosus* GG) groups. To induce colitis, 2% dextran sulfate sodium salt (DSS: molecular weight 36–50 kDa; MP biomedicals, Santa Ana, CA, USA) in the drinking water was administered to mice for 2 days, followed by 3% DSS administration to mice for 5 days. Mice were orally administered 2 × 10^8^ CFU/day of *L. sakei* SMFM2017-NK1 (LS1), *L. sakei* SMFM2017-NK3 (LS2), or *L. fermentum* SMFM2017-NK2 (LF), respectively, from the first day of DSS treatment. *L. rhamnosus* GG (LGG) was also administered, which is known to alleviate colitis, to compare its anti-inflammatory effect with other strains. PBS was administered to normal mice in the normal group and colitis-induced mice in the control group. While administering lactic acid bacteria, mice were fed a normal diet (Research diets, 173 Inc., New Brunswick, NJ, USA). After a two-week administration of lactic acid bacteria, mice were anesthetized by ether inhalation. This animal experiment was approved by the Institutional Animal Care and Use Committee (IACUC) of Sookmyung Women’s University (approval number: SMWU-IACUC-1805-008-01).

### 2.3. Weight Measurement

The body weights of mice were measured seven times throughout the entire experiment period. Weight gain rate was calculated as the ratio of the lowest weight after DSS administration to the maximum weight after oral administration of lactic acid bacteria.

### 2.4. Determination of Disease Activity Index (DAI)

Disease activity index (DAI) scores were graded every two days by observing the viscosity of feces and the presence or absence of blood in feces. DAI scores were calculated according to the DAI criteria [2] during adaptation and treatment periods.

### 2.5. Histopathological Analysis of Colon Tissue

Histopathological features of the colon were analyzed to compare the effects of different lactic acid bacteria on colitis. After anesthetizing mice, their colons were immediately extracted and rinsed with PBS. For pathological analysis of the colon, colon tissue was fixed with 10% formalin and embedded in paraffin, followed by sectioning to 3–4 µM in thickness. Sectioned tissues were stained with hematoxylin and eosin (H&E) and examined microscopically at magnifications of 50× and 200× by a board-certificated pathologist at the Republic of Korea Pathology Technical Center (KPNT, Cheongju, Chungcheongbuk-do, Republic of Korea). The degree and extent of inflammation and damage for each sample were observed under a microscope (Olympus bx50, Olympus Optical Co., Ltd., Tokyo, Japan), and scored according to a histologic colitis scoring system suggested by Erben et al. [10].

### 2.6. Analysis of Inflammation-Related Gene Expression in the Colon

mRNA was extracted from the colon using an RNeasy Lipid Tissue Mini kit (Qiagen, Hilden, Germany) according to the manufacturer’s instructions. In brief, complementary DNA (cDNA) was synthesized using the extracted mRNA, according to the manufacturer’s instructions of a QuantiTect Reverse Transcription kit (Qiagen). cDNA was then used to evaluate the gene expression of mRNA using a Rotor-Gene SYBR^®^ Green PCR kit (Qiagen). Quantitative reverse transcription-PCR (qRT-PCR) was performed using a Rotor-Gene Q (Qiagen). Gene expression levels of proinflammatory cytokines (TNF-α, IL-1β, IL-6, and IFN-γ) and mediators (iNOS and COX-2) were evaluated. In brief, a 25-μL reaction mixture contained 1 μL template cDNA, 12.5 μL 2× rotor-gene SYBR^®^ green PCR master mix, 6.5 μL RNase-free water, 2.5 μL forward primer, and 2.5 μL reverse primer. The following PCR conditions were used: initial denaturation at 95 °C for 5 min, followed by 35 cycles of 95 °C for 5 sec and 60 °C for 10 sec. Primer sequences used in this study are listed in Table 1. *β-actin *was used for the normalization of relative gene expression levels. Ct values were calculated by setting the threshold to 0.1. The relative gene expression was calculated with the 2^−ΔΔCt^ method [11].

### 2.7. Analysis of Biochemical Characteristics

Among the lactic acid bacteria evaluated, *L. fermentum* SMFM-NK2 was the most effective on colitis in the animal experiment based on the increase in colon length, the decrease in DAI score, and a low histopathological score. Therefore, the characteristics of this strain were further analyzed. Carbohydrate fermentation properties of *L. fermentum* SMFM2017-NK2 were analyzed using an API 50CHL kit (BioMérieux, Marcy, France) according to the manufacturer’s instructions. Briefly, 100 μL of the lactic acid bacteria strain in 20% glycerol stock was inoculated into 10 mL MRS broth and incubated at 37 °C for 24 h under anaerobic conditions prepared by anaerogen™ (Oxoid Ltd., Cheshire, UK) in a tightly sealed container. Afterward, 100 μL of the culture was transferred into fresh 10 mL MRS broth and incubated at 37 °C for 24 h. Then, 1 μL culture was streaked onto MRS agar and cultured at 37 °C for 24 h anaerobically. Several colonies on MRS agar were inoculated into 2 mL of sterile distilled water to obtain a turbid bacteria solution. The turbid bacterial solution was diluted in 5 mL of sterile distilled water to adjust the turbidity to 2 McFarland (BioMérieux). The finally adjusted bacterial solution was inoculated to 10 mL of API 50 CHL medium (BioMérieux). The medium with inoculum was then dispensed into API 50 CH strips (BioMérieux) and covered with mineral oil in strip tubes. After incubation at 37 °C for 24 h, the color change was observed for carbohydrate fermentation.

To analyze the catalase activity of lactic acid bacteria, the colony of lactic acid bacteria was mixed with 3% (*v/v*) hydrogen peroxide, and a reaction was observed based on the presence of air bubbles. If no air bubble was generated, the reaction was determined to be negative. To analyze the oxidase activities of lactic acid bacteria, the colony was swabbed directly on 1% N, N, N’, N’-tetramethyl-p-phenylenediamine strips (Microgen bioproducts Ltd., Camberley, UK), which were wetted with sterile distilled water. When the color of the strips turned dark blue or purple within 10 to 30 s, it was determined to be positive. If the color did not change, it was determined to be negative. The analysis of biochemical characteristics was performed with four replicates in each experiment.

### 2.8. Observation of Bacterial Morphology

A single colony of *L. fermentum* SMFM2017-NK2 was inoculated into MRS broth and incubated anaerobically at 37 °C for 24 h. The culture (100 μL) was inoculated into 10 mL fresh MRS broth containing sterile glass (0.5 × 0.5 cm) and incubated anaerobically at 37 °C for 24 h to allow bacteria to attach to the glass. The glass was then transferred to a 24-well plate and 50 μL of 1.8% glutaraldehyde solution (Sigma, St. Louis, MO, USA) was added. The glass was washed with 1 mL sterile distilled water three times for 5 min. After washing, the moisture from the glass was removed with sterile gauze, and 40 μL of 2% osmium tetroxide solution (Sigma) was dropped onto the glass for secondary fixation at room temperature in a dark condition for 20 min. The glass was then washed with 1 mL sterile distilled water three times for 5 min and soaked with 25, 50, 75, 90, and 100% ethanol for 5 min each. The glass was dried at room temperature for 1 h. Then, 40 μL of hexamethyldisilazane (HMDS; Sigma) was added to dry it completely. The surface of the glass was coated with platinum using a Sputter Coater (Cressington Ltd., Oxhey, Watford, UK) to give conductivity. It was fixed to the mount, and the morphology of the bacteria was observed with a field emission scanning electron microscope (FE-SEM; JEOL USA Inc., Peabody, MA, USA).

### 2.9. Analysis of Cytotoxicity

HT-29 cells (human colon cancer cells, KCLB30038) were purchased from Republic of Korea Cell Line Bank (Seoul, Republic of Korea) and used for analyzing the cytotoxicity of *L. fermentum* SMFM2017-NK2. Briefly, HT-29 cells were cultured in Dulbecco’s Modified Eagles Medium (DMEM; Hyclone, Logan, UT, USA) supplemented with 10% fetal bovine serum (FBS; Hyclone) and 1% penicillin-streptomycin solution (PS; Oxoid Ltd.) in 72T flasks (Corning, Oneonta, NY, USA), and incubated at 37 °C under 5% CO_2_. Cultured cells were then sub-cultured in the fresh medium for 2 days and washed with Dulbecco’s Phosphate-Buffered Saline (DPBS; Welgene, Gyeongsan, Gyeongsangbuk-do, Republic of Korea). These cells were then detached with 0.05% trypsin-0.02% EDTA (Gibco, Grand Island, NY, USA). Separated cells were centrifuged at 1000 rpm and 25 °C for 5 min. Then, 5 × 10^4^ cells/mL were seeded into a 96-well microplate (SPL Life Sciences, Gyeonggi-do, Republic of Korea) and incubated at 37 °C with 5% CO_2_ for 24 h. After removing the supernatant, 200 μL of lactic acid bacteria at 6, 7, and 8 Log CFU/mL in DMEM without antibiotics was added to the wells. The cells treated with PBS and DMEM were used as controls. Afterward, the supernatant was carefully removed, and gentamicin (50 μg/mL) was used for treatment for 2 h to remove any remaining bacteria in the cells. After 3-(4,5-Dimethylthiazol-2-yl)-2,5-diphenyltetrazolium bromide (MTT; Sigma) reagent was added to each well, the cells were incubated at 37 °C for 2 h to metabolize MTT. After removing the reagent and dissolving the formazan crystal using DMSO, absorbance was measured at 540 nm in a microplate spectrophotometer (Take3, Epoch, BioTek, Winooski, VT, USA). The cytotoxicity was calculated following the manufacturer’s instructions (MTT; Sigma) with a slight modification:Cytotoxicity (%) = Absorbance of sample/absorbance of control × 100

### 2.10. Evaluation of Heat Stability

First, 100 μL *L. fermentum* SMFM2017-NK2 in 20% glycerol stock was inoculated in 10 mL of MRS broth and incubated at 37 °C for 24 h under an anaerobic condition, which was prepared with anaerogen™ in a tightly sealed container. Afterward, 100 μL of the cultures was transferred into fresh 10 mL MRS broth and incubated at 37 °C for 24 h. Then, 1.0 × 10^9^ CFU/mL of lactic acid bacteria (100 μL) was inoculated into 10 mL MRS broth and exposed to 55 °C for 6 h. After 0.1 mL of the culture was retrieved at 1, 3, and 6 h, cultures were diluted with 0.1% buffered peptone water (BPW; Becton, Dickinson and Company). Then, 0.1 mL diluents were plated onto MRS agar and incubated at 37 °C for 24 h anaerobically. The number of viable cells was counted, and bacterial viability was calculated as follows:

Reduction of viable bacteria = log N_0_ − log N

N_0_: initial cell count

N: viable count after heat treatment

### 2.11. DNA Library Preparation and Sequencing

To analyze the genomic characteristics of *L. fermentum* SMFM-NK2, whole genome *de novo* sequencing was performed. The DNA of *L. fermentum* SMFM-NK2 was prepared using a TruSeq Nano DNA Sample Preparation Kit (Illumina, San Diego, CA, USA) according to the manufacturer’s procedure. In brief, genomic DNA was fragmented to construct a library. Blunt-end fragments were created and adenylated with A-base. Dual-index adapters were ligated to DNA fragments and then amplified for generating clusters. Sequencing was performed with the method of sequencing by synthesis, which used four fluorescently labeled nucleotides to sequence clusters on the surface of the flow cell in parallel with a HiSeq 2500 system (Illumina) [18]. The quantity and purity of DNAs were checked with a spectrophotometer (Infinium F-200, NanoDrop, Illumina) and a fluorometer (Qubit, Life Technology, Carlsbad, CA, USA). The quality (per base sequence quality, per tile sequence quality, per sequence quality scores, and per base sequence content) of raw sequencing data was visualized using FastQC software (Illumina). Whole metagenome *de novo* assembly was performed using a *de novo* assembler IDBA-UD algorithm [19].

### 2.12. Gene Prediction and Taxonomic Profiling

The assembled genome sequence of *L. fermentum* SMFM-NK2 was annotated and predicted using Prokka ver.1.10, a software tool fully annotating a draft bacterial genome [20], relying on the UniProt for protein information, RefSeq for nucleotide sequences, and Pfam for protein family databases [21,22,23,24]. Taxonomic profiling of the aligned genome was performed using NCBI taxonomy information and Krona tools [25].

### 2.13. Orthologous Genes Clustering

Whole genome sequences of *L. fermentum* strains F-6, LDTM 7301, and MTCC 25067 were used to compare with whole genome sequences of *L. fermentum* SMFM-NK2. Whole genome sequences were downloaded from GenBank of NCBI (National Center for Biotechnology Information) database. Genes of *L. fermentum* species, which were collected from the results of taxonomy profiling, were used for clustering orthologous genes using an OrthoMCL program [26].

### 2.14. Analysis of Single Nucleotide Polymorphism (SNP) and Insertion/Deletion (InDel)

Reads of *L. fermentum* SMFM2017-NK2 were aligned to *L. fermentum* strain LDTM 7301 chromosome (complete genome) as a reference genome, which had the most similar homology to *L. fermentum* SMFM2017-NK2. Genetic variants, which mean differences between the genome of *L. fermentum* SMFM2017-NK2 and the reference genome, were annotated using the SnpEff tool [27]. These variants were described as single nucleotide polymorphism (SNP), insertion (Ins), and deletion (Del).

### 2.15. Statistical Analysis

Data on body weights and gene expression levels were analyzed by the general linear model (GLM) procedure of SAS^®^ version 9.4 (SAS Institute, Inc., Cary, NC, USA). The significance was determined with a pairwise *t*-test at *α =* 0.05. Values of the normal group were compared to those of the control group. Values of the lactic acid bacteria-administered groups were also compared to those of the control group.

## 3. Results and Discussion

### 3.1. Body Weight

Weight loss is known to be a major feature that appears when inflammation is induced by the administration of DSS [28]. Weights of mice in control, LS1, LS2, LF, and LGG groups, which were DSS-treated for 7 days (2 days for 2% DSS and 5 days for 3% DSS), were decreased sharply on days 5–11. After oral administration of lactic acid bacteria, the weights of mice were increased gradually from day 11. The weight of the LF group was the highest among the other treated groups. When the lowest weight after DSS administration and the maximum weight after administration of lactic acid bacteria were compared, the LF group showed the highest weight increase rate at 8.91% (Figure 1). The control group showed a weight gain of 6.5% after stopping DSS, which was described similarly in a study by Ahn et al. [29].

### 3.2. Length of Large and Small Intestine

Severe inflammation affects the lengths of the large and small intestines and the length of the large intestine of mice with inflammation compared to normal mice [30]. In our study, the lengths of the large (4.7 cm) and small intestines (34.5 cm) in the control group were slightly different from those of the normal group (large intestine: 4.9 cm; small intestine: 35.2 cm). The lengths of the large intestine for LF (5.2 cm), LS2 (4.8 cm), and LGG (4.9 cm) groups were slightly longer than that of the control group (4.7 cm). The length of the small intestine of the LF (38 cm) group was the longest (*p* < 0.05) compared to those of the control (34.5 cm), LS1 (36 cm), LS2 (34.3 cm), and LGG (34.8 cm) groups. This result indicates that DSS treatment could decrease the colon lengths of mice. However, colon length increased after administration of *L. fermentum* SMFM2017-NK2. Ahn et al. [29] have also shown that the length of the colon increased after ingesting a mixture of *Lactobacillus brevis* HY7401, *Lactobacillus helveticus* HY7801, and *Bifidobacterium longum* HY8004 in 2.5% (*w/v*) DSS-administered mice.

### 3.3. DAI Score during Administration of Lactic Acid Bacteria

The DSS colitis model has clinical signs characterized by weight loss, diarrhea, and bloody stools [31]. Indicators of disease activity can be expressed through these clinical signs. In our studies, the conditions of feces were checked every two days and the degree of inflammation was evaluated according to disease activity index (DAI) criteria [2]. On day 3, loose stools were observed in the control group and the LF group (score 0.5). On day 5 (score 3.5) and day 7 (score 3), loose stools and bloody stools were observed in all mice except for those of the normal group. From day 7 when DSS administration was discontinued and only oral administration of lactic acid bacteria was performed, watery stool decreased in control and lactic acid bacteria-administered groups (LS1, LS2, LF, and LGG group). In addition, watery stool and bloody stool severity decreased on day 9 (score 1.75). On day 11, the DAI score was 1 for both the control and LS1 groups, and the DAI score was 0.5 for the LS2, LF, and LGG groups.

### 3.4. Histopathological Features of Colon

When inflamed in the large intestine, the mucous membrane of the large intestine is known to show infiltration and ulcers. In severe cases, the colonic mucosa produces large erosion, and the mucus is destroyed. The mucous membrane layer becomes thinner, with local congestion, edema, and fibrosis seen under the mucous membrane [32]. In the LS1 group, inflammatory cell infiltration was observed in the muscle layer of the colon. Infiltration of inflammatory cells, loss of crypt, cell debris in the lumen, and edema of inflammatory cells and submucosal tissues were also observed (Figure 2A). In the LS2 group, inflammatory cell infiltration and intestinal gland loss were observed in the colonic mucosa (Figure 2B). In the LF group, inflammatory cell infiltration was observed in the mucous membrane layer of the colon (Figure 2C). In the LGG group, inflammatory cell infiltration was observed in the serous layer of the colon (Table 2, Figure 2D). When tissue lesions were compared according to the histologic colitis scoring system [10], the criteria could categorize lesions into six groups (0–3 grade of the score for each category): infiltration of inflammatory cells, depth of inflammatory cell infiltration, crypt, edema, mucoprotein and hemorrhage, and pseudomembrane. Total scores for lesions were compared among groups. The LS1 group had a score of 23. However, the lesions were low in the LF (score of 8) and LS2 (score of 13) groups. In particular, LF (score of 8) was the most effective in alleviating inflammation in the colon. The total score of the LGG group was 16, which was less effective than LF.

It is known that when DSS is administered to mice, it is directly toxic to intestinal epithelial cells due to inflammation, which affects the integrity of the mucosal cavity [29]. Thus, the DSS-induced model is generally used for inducing colitis. Negatively charged DSS can destroy colonic epithelia and increase epithelial permeability, specifically in the distal colon. Major characteristics of colitis are weight loss, watery stool, and stool bleeding. Acute or chronic colitis is induced by adjusting DSS concentration and/or frequency. The DSS-induced model is similar to UC in humans. Thus, many researchers have used this model for IBD [33].

According to previous studies, *Lactobacillus brevis* HY7401, *L. helveticus* HY7801, and *Bifidobacterium longum* HY8004 could alleviate the destruction of colon epithelial cells caused by colitis in a mouse model [29]. In our study, lactic acid bacteria isolated from kimchi also reduced inflammation, especially *L. fermentum* SMFM2017-NK2 and *L. sakei* SMFM2017-NK3, which had higher effects than LGG.

LS1 (colitis-induced mice administered with *L. sakei* SMFM2017-NK1), LS2 (colitis-induced mice administered with *L. sakei* SMFM2017-NK3), LF (colitis-induced mice administered with *L. fermentum* SMFM2017-NK2), and LGG (colitis-induced mice administered with *L. rhamnosus* GG) groups. Magnification of 50× (above) and 200× (below).

### 3.5. Gene Expression of Proinflammatory Cytokines in Colon

In this study, gene expression levels of proinflammatory cytokines (TNF-α, IL-1β, IL-6, and IFN-γ) and inflammation-related mediators (iNOS and COX-2) were analyzed to confirm the inflammatory response. The gene expression level of TNF-α was higher in the control group than that in the normal group. After treatment with lactic acid bacteria, the gene expression level of TNF-α was not decreased compared to that in the control group. However, the gene expression level of TNF-α was the lowest in the LF group among the lactic acid bacteria-treated groups (Figure 3A). Regarding IL-1β, the gene expression level was higher in the control than that in the normal group (Figure 3B). The LF group had the lowest IL-1β gene expression level among the other groups (Figure 3B). The gene expression level of IL-6 was higher (*p* < 0.05) in the control than in the normal group. Gene expression levels in LS1, LS2, LF, and LGG groups were lower than those in the control group. The LF group was the lowest among lactic acid bacteria-treated groups (Figure 3C). The gene expression level of IFN-γ in the normal group was higher than that of the control group (Figure 3D). Gene expression of iNOS known to produce NO and was higher in the control group than in the normal group (Figure 3E). Its level in the LF group was the lowest among all lactic acid bacteria-treated groups (Figure 3E). The expression level of COX-2 was higher in the control group than in the normal group. There was no significant difference between the lactic acid bacteria-treated groups and the control group (Figure 3F). These results indicate that *L. fermentum* SMFM2017-NK2 (LF) could alleviate inflammation in the colon by decreasing gene expression levels of pro-inflammatory cytokine IL-6 compared to other lactic acid bacteria strains.

Several *L. fermentum* strains are effective on ulcerative colitis by regulating inflammatory cytokines. *L. fermentum* ZS40 could down-regulate gene expression of IL-6 and TNF-α [34]. *L. fermentum* CQPC04 could inhibit the release of pro-inflammatory cytokines such as TNF-α, IFN-γ, IL-1β, IL-6, and IL-12 [35]. *L. fermentum* HFY06 could inhibit the release of cytokines TNF-α, iNOS, and COX-2 [36]. Similar to these research studies, we focused on the mechanism of decreasing pro-inflammatory cytokines, especially IL-6.

### 3.6. Biochemical Characteristics

Lactic acid bacteria use a carbon source such as sugar for proliferation [37]. They are useful bacteria with desirable effects on the human body by producing large amounts of organic acids, especially lactic acid, using a carbon source such as glucose or lactose. For mass production of probiotic strains, it is first necessary to establish conditions of the medium to be suitable for each strain. Since nutritional requirements and growth characteristics required for each strain are diverse, the types and concentrations of carbon sources, nitrogen sources, and trace elements for each strain should be optimized [38]. Thus, it is necessary to check which carbon source is being used. Therefore, our study evaluated the carbon availability of lactic acid bacteria. In our study, *L. fermentum* SMFM2017-NK2 fermented 27 carbohydrates (glycerol, L-arabinose, ribose, D-galactose, D-glucose, D-fructose, D-mannose, D-rhamnose, mannitol, sorbitol, N-acetyl-glucosamine, amygdaline, arbutin, esculin, salicin, cellobiose, maltose, lactose, melibiose, sucrose, trehalose, inuline, melezitose, raffinose, gentiobiose, D-turanose, and gluconate) without fermenting 22 carbohydrates (erythritol, D-arabinose, D-xylose, L-xylose, D-adonitol, Metil-BD-xylopyranosicle, D-sorbose, dulcitol, inositol, Metil-D-mannoside, Metil-D-glucoside, starch, glycogen, xylitol, D-lyxose, D-tagatose, D-fucose, L-fucose, D-arabitol, L-arabitol, 2-kero-gluconate, 5-kero-gluconate).

In general, lactic acid bacteria are anaerobic bacteria that grow well without being affected by oxygen [1]. *Lactobacillus* refers to catalase-negative anaerobic bacteria [39]. Catalase is an enzyme involved in decomposing H_2_O_2_ into H_2_O and O_2_ [40]. Bacteria reacting with oxidase negative do not have cytochrome c oxidase. Thus, they may respire using other oxidases in electron transport [41]. Results for catalase and oxidase activity of *L. fermentum* SMFM2017-NK2 were all negative, indicating that *L. fermentum* SMFM2017-NK2 is anaerobic bacteria.

### 3.7. Morphological Characteristics

When lactic acid bacteria are used as probiotics, precise characteristics are required. Thus, the morphology of lactic acid bacteria was observed. *L. fermentum* SMFM2017-NK2 was rod-shaped without motility, as seen in the scanning electron microscope (Figure 4). Its characteristics were similar to the characteristics of a typical strain of the genus *Lactobacillus*, a form of gram-positive bacilli [42].

### 3.8. Cytotoxicity

It is not suitable to use cytotoxic lactic acid bacteria to develop functional foods or drugs. Thus, cytotoxicity should be evaluated. HT-29 cells are similar to human epithelial cells, which are proper for analyzing cytotoxicity [43]. Cell toxicity can be measured with MTT assay using cellular protein staining and measuring cell viability [44]. In this study, the cell viability of *L. fermentum* SMFM2017-NK2 was 119.6% at 8 Log CFU/mL, 91.6% at 7 Log CFU/mL, and 93.3% at 6 Log CFU/mL. Cell viability of LGG was 95.9% at 8 Log CFU/mL, 87.5% at 7 Log CFU/mL, and 107.1% at 6 Log CFU/mL. Except for the low inoculation concentration (6 Log CFU/mL) of lactic acid bacteria, cell viability was higher in *L. fermentum* SMFM2017-NK2 than in LGG. This result indicates that *L. fermentum* SMFM2017-NK2 might not have cytotoxicity.

### 3.9. Heat Stability

Maintaining viable cell counts of lactic acid bacteria is essential for the probiotic pelleting process [45]. Thus, it is necessary to select strains with high resistance to heat. In this study, strains were analyzed for heat stability. After heating at 55 °C for 6 h, *L. fermentum* SMFM2017-NK2 had a death rate of 20.2%, which was higher than that of LGG at 11.9%. Although the LGG strain had a lower death rate, *L. fermentum* SMFM2017-NK2 was more stable than the other *L. fermentum* isolates obtained from our laboratory (*L. fermentum* SMFM2017-NK1: 27% of death rate, *L. fermentum* SMFM2017-NK3: 24.8% of death rate).

### 3.10. General Features of L. fermentum SMFM2017-NK2 Genome

The whole genome of *L. fermentum* SMFM2017-NK2 was sequenced. The generated library yielded 9,243,170 reads with an average length of 101 bp. The GC content and size of *L. fermentum* SMFM2017-NK2 were 50.5% and 0.93 Gb, respectively. Q30 (99.9% accuracy of sequence) and Q20 (99% accuracy of sequence) were 93.3% and 96.1%, respectively, indicating the good quality of the library. After whole genome *de novo* assembly for *L. fermentum* SMFM2017-NK2, 1,096 scaffolds were obtained and 5,682 CDS, 114 tRNA, 2 RNA, and 5 repeat regions were predicted. Through taxonomic annotation for scaffolds, *L. fermentum* SMFM2017-NK2 was close to *L. fermentum* with 92.4% scaffolds read ratio (Figure 5).

### 3.11. Orthologous Gene Cluster of L. fermentum SMFM2017-NK2

Orthologs are genes in different species that have evolved from a single ancestral gene [46]. They were identified between *L. fermentum* SMFM2017-NK2 and three other *L. fermentum* strains. Sequences or functions of these genes are similar if genes are in the same orthologous group. Results of taxonomic profiling revealed that *L. fermentum* SMFM2017-NK2, *L. fermentum* F6, *L. fermentum* LDTM 7301, and *L. fermentum* MTCC 25067 had 1966, 2006, 1834, and 2020 genes, respectively. Through this gene information, 1875 orthologous groups were identified. All four *L. fermentum* species shared 1385 orthologous groups. *L. fermentum* SMFM2017-NK2 shared 1613, 1635, and 1491 orthologous groups with *L. fermentum* F6, *L. fermentum* LDTM 7301, and *L. fermentum* MTCC 25067, respectively (Figure 6). Seven orthologous groups (H001716, H001717, H001737, H001777, H001778, H001789, and H001794 groups) were identified only in *L. fermentum* SMFM2017-NK2, which could be differentiated from the other three *L. fermentum* species (Table 3). There are 14 genes in seven orthologous groups (Table 3). Of 14 genes, 11 genes (GENE_01478, GENE_02186, GENE_01485, GENE_02175, GENE_02180, GENE_04647, GENE_03909, GENE_04681, GENE_03910, GENE_04624, and GENE_050499) were annotated as hypothetical proteins, which were predicted to be expressed without experimental evidence [47]. Two of the genes (GENE_05028 and GENE_05029) were annotated as OsmC-like proteins, and GENE_04680 was annotated as a PD-(D/E)XK nuclease superfamily protein. PD-(D/E)XK nuclease superfamily protein is related to DNA recombination and repair [48]. OsmC is an osmotically inducible protein that encodes an envelope protein [49]. These proteins can form characteristics of *L. fermentum* SMFM2017-NK2.

### 3.12. Genomic Variation of L. fermentum SMFM2017-NK2

Genomic variants comprised single nucleotide polymorphism (SNP) and insertion and deletion (InDel). These variants of *L. fermentum* SMFM2017-NK2 were analyzed and compared to the genome of *L. fermentum* strain LDTM 7301, which was the most similar to *L. fermentum* SMFM2017-NK2 analyzed by orthologous gene cluster. It was found that 7.55% of single nucleotide variants were present in the exon region (Table 4). The number of SNPs was 3055, indicating 2743 homozygous and 312 heterozygous SNPs. The number of InDels was 181, consisting of 149 homozygous and 32 heterozygous InDels. Effects of variants were categorized as high (92 variants), low (1561 variants), moderate (989 variants), and modifier (32,428 variants) by the impact in reference to the standard of SnpEff (https://pcingola.github.io/SnpEff/se_introduction/ (accessed on 29 December 2020)) [50]. The 92 variants in the high category might have a high (disruptive) impact on the protein, such as causing protein truncation or loss of function. The predicted effect was a frameshift caused by insertion or deletion (‘frameshift_variant’; 49 variants), causing a stop codon (‘stop_gained’; 19 variants), or causing a stop codon to be mutated into a non-stop codon (‘stop_lost’; 8 variants) (Table 5). In addition, ‘splice_region_variant’ means a variant effective putative (Lariat) branch point from U12 splicing machinery located in the intron, ‘inframe_deletion’ means that one or many codons is(are) deleted, and ‘disruptive_inframe_insertion’ means one codon is changed and one or many codons is(are) inserted. These frameshifts by insertion or deletion were found in several variants (‘frameshift_variant & stop_gained’, eight variants; ‘frameshift_variant & stop_lost’, two variants; ‘start_lost & inframe_deletion’, 1 variant; ‘stop_gained & disruptive_inframe_insertion’, one variant; ‘stop_lost & splice_region_variant’, four variants) (Table 5). These variants could affect the characteristics of *L. fermentum* SMFM2017-NK2 in gene expression, different from those of other *L. fermentum* strains.

## 4. Conclusions

The length of the colon, DAI, histopathologic score, and inflammation-related gene expression in the LF group indicated that *L. fermentum* SMFM2017-NK2 could effectively alleviate colitis. Biochemical characteristics, cytotoxicity, and heat stability of *L. fermentum* SMFM2017-NK2 suggest that it could be a probiotic strain. In addition, comparative genomic analysis results indicate that *L. fermentum* SMFM2017-NK2 is a novel strain. Therefore, *L. fermentum* SMFM2017-NK2 might be appropriate for anti-inflammation in the colon. Limitations of this study include the lack of sample size in the animal experiments, results only from the animal study, and the lack of various results related to immune response or gut barrier functions. Thus, further studies should be performed to clarify the mode of action for the anti-inflammatory effects of *L. fermentum* SMFM2017-NK2 on the colon. In addition, probiotic safety is a prerequisite for human use to avoid health risks. Safety evaluation of probiotics has been presented in the FAO/WHO guidelines (2002) [51]. In brief, antibiotic resistance of probiotics is one of the primary safety issues because antibiotics cannot be affected if probiotics with antibiotic resistance genes are present in the gut. D-lactate acidosis can occur in patients with short bowel syndrome. It is accompanied by cognitive and neurological impairments [52]. In addition, deconjugated bile salts can produce secondary metabolites such as glycine and taurine, which can be toxic [53]. Therefore, safety evaluation, such as antibiotics resistance, presence of toxic genes, production of D-lactate, bile salt deconjugation, and so on should be examined for *L. fermentum* SMFM2017-NK2

## Figures and Tables

**Figure 1 microorganisms-11-00547-f001:**
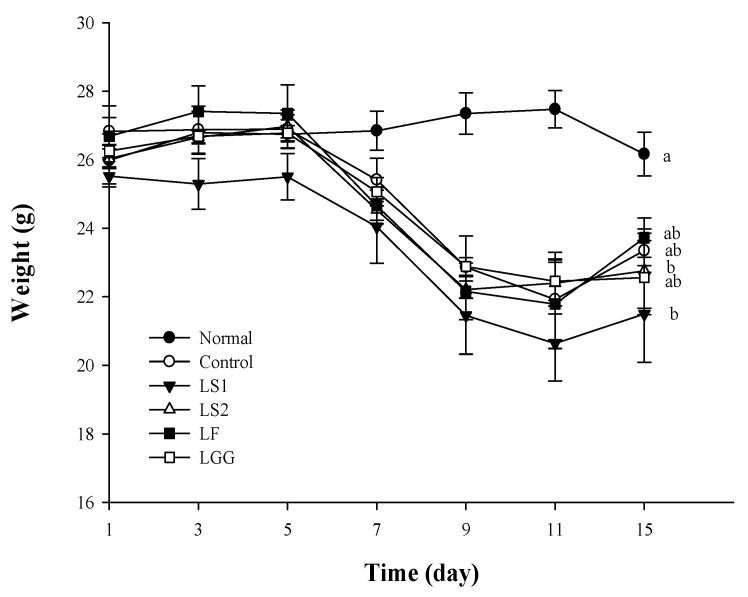
Body weight change of mice. Normal (normal mice administered with PBS), control (colitis-induced mice administered with PBS), LS1 (colitis-induced mice administered with *L. sakei* SMFM2017-NK1), LS2 (colitis-induced mice administered with *L. sakei* SMFM2017-NK3), LF (colitis-induced mice administered with *L. fermentum* SMFM2017-NK2), and LGG (colitis-induced mice administered with *L. rhamnosus* GG) groups. a,b; means in figure with different letters are significantly different (*p* < 0.05).

**Figure 2 microorganisms-11-00547-f002:**
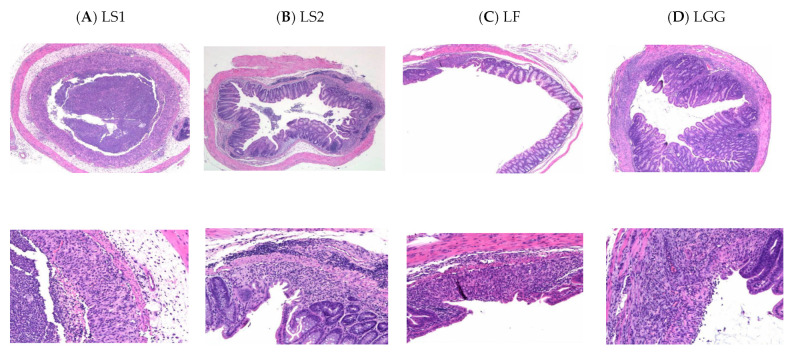
Histopathological features of the colon.

**Figure 3 microorganisms-11-00547-f003:**
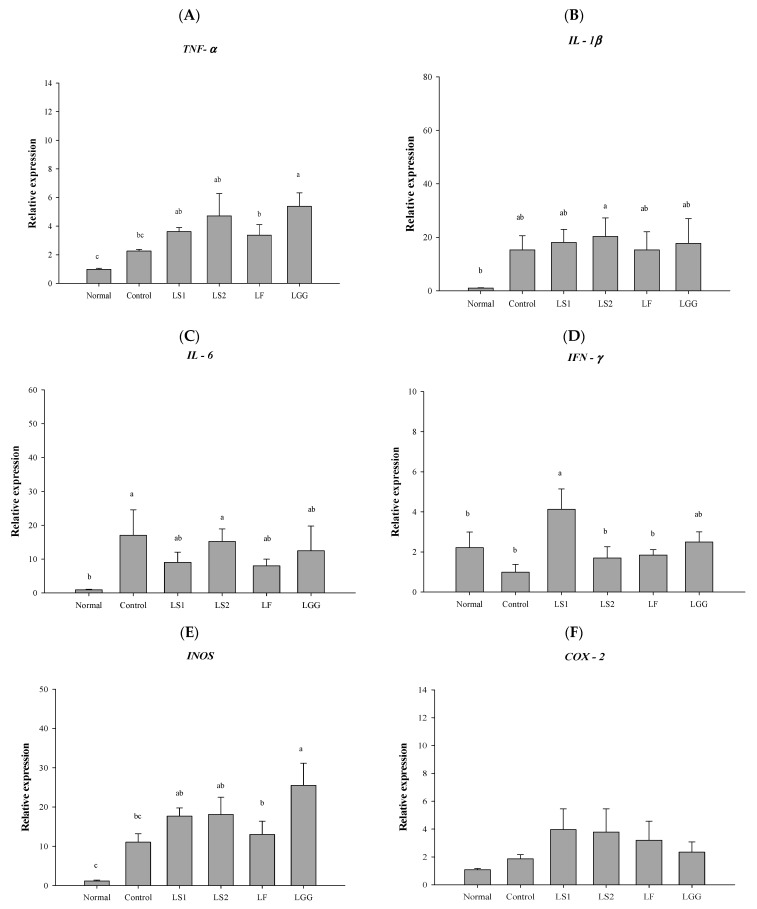
Relative gene expression levels of inflammatory cytokine [TNF-α (**A**), IL-1β (**B**), IL-6 (**C**), IFN-γ (**D**), iNOS (**E**), and COX-2 (**F**)] in colon. Normal (normal mice administered with PBS), control (colitis-induced mice administered with PBS), LS1 (colitis-induced mice administered with *L. sakei* SMFM2017-NK1), LS2 (colitis-induced mice administered with *L. sakei* SMFM2017-NK3), LF (colitis-induced mice administered with *L. fermentum* SMFM2017-NK2), and LGG (colitis-induced mice administered with *L. rhamnosus* GG) groups. a–c; means in figure with different letters are significantly different (*p* < 0.05).

**Figure 4 microorganisms-11-00547-f004:**
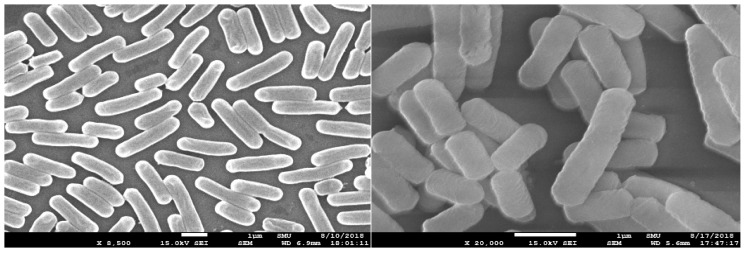
Morphology of *L. fermentum* SMFM2017-NK2 by field emission scanning electron microscope.

**Figure 5 microorganisms-11-00547-f005:**
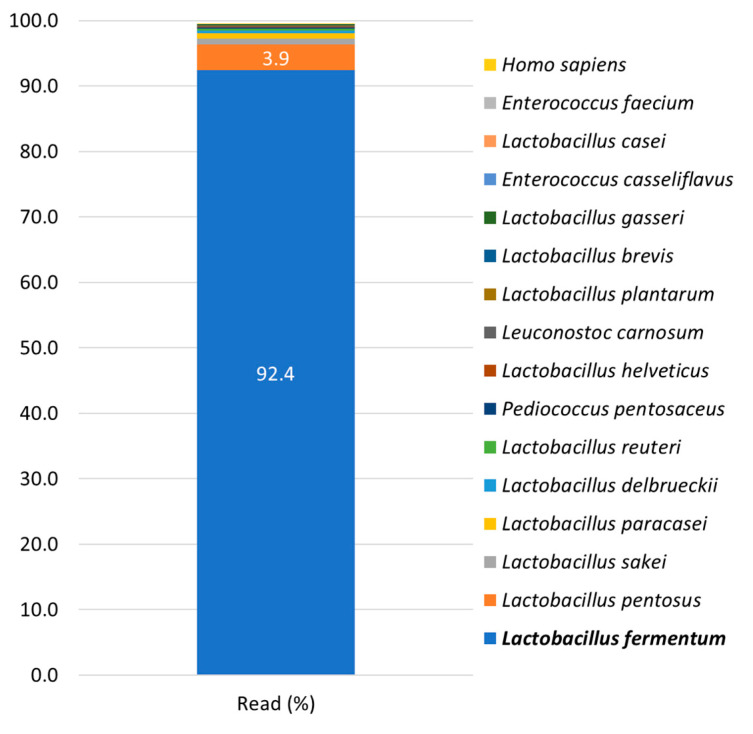
Taxonomy profiling of *L. fermentum* SMFM2017-NK2 by whole metagenome sequencing.

**Figure 6 microorganisms-11-00547-f006:**
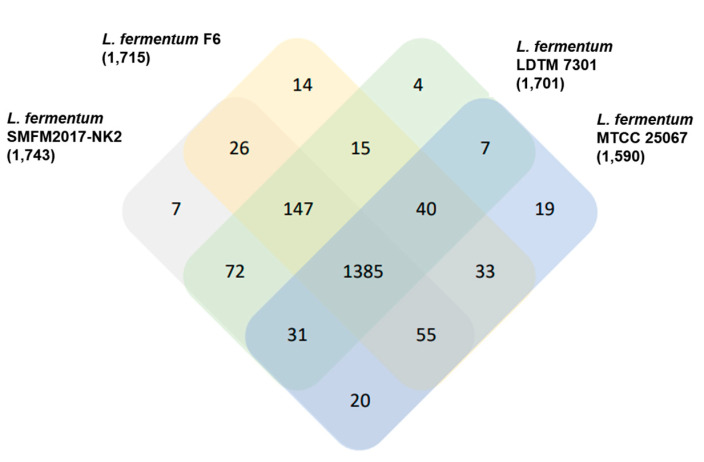
Clusters of orthologous groups (COG) among *L. fermentum* SMFM2017-NK2, *L. fermentum* F6, *L. fermentum* LDTM 7301, and *L. fermentum* MTCC 25067.

**Table 1 microorganisms-11-00547-t001:** Primer sequences of inflammation-related genes used for the real-time PCR.

Gene		Primer Sequence (5′->3′)	References
β-actin	Forward	CCGTGAAAAGATGACCCAGATC	[12]
	Reverse	CACAGCCTGGATGGCTACGT	
TNF-α	Forward	TCTTCTCATTCCTGCTTGTGG	[13]
	Reverse	GGTCTGGGGCATAGAAGGA	
IL-1β	Forward	AACCTGCTGGTGTGTGACGTTC	[12]
	Reverse	CAGCACGAGGCTTTTTTGTTGT	
IL-6	Forward	ACCAGAGGAAATTTTGAATAGGC	[14]
	Reverse	TGATGCACTTGCAGAAAACA	
INOS	Forward	AATCTTGGAGCGAGTTGTGG	[15]
	Reverse	CAGGAAGTAGGGAGGGCTTG	
IFN-γ	Forward	AGCGGCTGACTGAACTCAGATTGTAG	[16]
	Reverse	GTCACAGTTTTCAGCTGTATAGGG	
COX-2	Forward	TGTATCCCCCCACAGTCAAAGACAC	[17]
	Reverse	GTGCTCCCGAAGCCAGATGG	

**Table 2 microorganisms-11-00547-t002:** Histopathological score of colon after administration of lactic acid bacteria.

Histopathological Features	Group															
	LS1				LS2				LF				LGG			
Infiltration of inflammatory cells	3	3	3	0	3	2	1	0	1	3	0	0	0	1	3	3
Depth of inflammatory cell infiltration	2	2	2	0	1	1	1	0	1	1	0	0	0	1	1	3
Crypt	2	2	3	0	1	2	0	0	1	1	0	0	0	0	1	2
Edema	0	0	1	0	1	0	0	0	0	0	0	0	0	0	0	1
Mucoprotein	0	0	0	0	0	0	0	0	0	0	0	0	0	0	0	0
Hemorrhage	0	0	0	0	0	0	0	0	0	0	0	0	0	0	0	0
Pseudomembrane	0	0	0	0	0	0	0	0	0	0	0	0	0	0	0	0
Sum	23				13				8				16			

LS1 (colitis-induced mice administered with *L. sakei* SMFM2017-NK1), LS2 (colitis-induced mice administered with *L. sakei* SMFM2017-NK3), LF (colitis-induced mice administered with *L. fermentum* SMFM2017-NK2), and LGG (colitis-induced mice administered with *L. rhamnosus* GG) groups.

**Table 3 microorganisms-11-00547-t003:** Functional cluster of orthologous groups only in *L. fermentum* SMFM2017-NK2 genome not existing in the genome of other *L. fermentum* strains (*L. fermentum* F6, *L. fermentum* LDTM 7301, and *L. fermentum* MTCC 25067).

Orthologous Group	Gene	Start Position	End Position	Sequence Length (bp)	Type	Function
H001716	GENE_01478	382	591	210	CDS	hypothetical protein
	GENE_02186	160,903	161,112	210	CDS	hypothetical protein
H001717	GENE_01485	3011	3388	378	CDS	hypothetical protein
	GENE_02175	156,223	156,648	426	CDS	hypothetical protein
H001737	GENE_02180	159,325	159,477	153	CDS	hypothetical protein
	GENE_04647	106,749	106,883	135	CDS	hypothetical protein
H001777	GENE_03909	32	433	402	CDS	hypothetical protein
	GENE_04681	120,449	120,838	390	CDS	hypothetical protein
H001778	GENE_03910	426	1274	849	CDS	hypothetical protein
	GENE_04680	119,608	120,456	849	CDS	PD-(D/E)XK nuclease superfamily protein
H001789	GENE_04624	78,078	78,485	408	CDS	hypothetical protein
	GENE_05049	900	1265	366	CDS	hypothetical protein
H001794	GENE_05028	1293	1730	438	CDS	OsmC-like protein
	GENE_05029	1742	2164	423	CDS	OsmC-like protein

**Table 4 microorganisms-11-00547-t004:** Region of the single nucleotide variants of *L. fermentum* SMFM2017-NK2 compared with *L. fermentum* strain LDTM 7301.

Region of the Genome	Count	Percent (%)
Upstream	15,522	44.26
Downstream	16,302	46.48
Exon	2648	7.55
Intergenic	596	1.70
Transcript	1	0.00
None	1	0.00

**Table 5 microorganisms-11-00547-t005:** Effect of single nucleotide variants of *L. fermentum* SMFM2017-NK2 compared with *L. fermentum* strain LDTM 7301.

Effect	Position	Reference Allel	Alternative Allele	Depth	Codon	Amino Acid	Gene	Transcript
Frameshift_variant	16,086	C	CG	369	c.635dupG	p.Val213fs	Gene_gene17	gene17
	135,033	C	CG	391	c.96dupG	p.Pro33fs	Gene_gene151	gene151
	157,379	T	TG	319	c.516dupG	p.Arg173fs	Gene_gene175	gene175
	327,600	A	AC	216	c.538dupC	p.Arg180fs	BGV76_RS01760	Transcript_gene350
	401,859	C	CA	716	c.524dupT	p.Ile176fs	BGV76_RS02115	Transcript_gene421
	402,254	A	ATCGC	580	c.129_130insGCGA	p.Tyr44fs	BGV76_RS02115	Transcript_gene421
	402,280	GA	G	638	c.103delT	p.Ser35fs	BGV76_RS02115	Transcript_gene421
	531,259	CT	C	308	c.209delA	p.Glu70fs	Gene_gene559	gene559
	579,296	G	GC	288	c.126dupC	p.Ser43fs	Gene_gene613	gene613
	606,522	C	CG	321	c.968dupG	p.Cys324fs	Gene_gene637	gene637
	619,737	A	AAT	59	c.4_5insAT	p.Ser2fs	Gene_gene646	gene646
	664,789	T	TCCTAAATTGCAAGATTAAGTGAGCCACCCGGCCACGGGAG	93	c.410_411insCTCCCGTGGCCGGGTGGCTCACTTAATCTTGCAATTTAGG	p.Gln137fs	tnpA	Transcript_gene696
	727,563	T	TC	274	c.557dupG	p.Val187fs	Gene_gene763	gene763
	881,163	G	GC	311	c.698dupG	p.Lys234fs	Gene_gene905	gene905
	1,039,055	AG	A	427	c.1167delC	p.Ser389fs	BGV76_RS05400	Transcript_gene1078
	1,076,406	CG	C	408	c.383delG	p.Gly128fs	BGV76_RS05580	Transcript_gene1114
	1,076,406	CG	C	408	c.19delG	p.Ala7fs	BGV76_RS05585	Transcript_gene1115
	1,094,586	CG	C	259	c.75delC	p.Cys25fs	BGV76_RS05690	Transcript_gene1136
	1,094,590	CA	C	264	c.71delT	p.Val24fs	BGV76_RS05690	Transcript_gene1136
	1,212,247	AACAAAGAAAT	A	63	c.242_251delACAAAGAAAT	p.Asn81fs	Gene_gene1265	gene1265
	1,212,259	CTATT	C	50	c.254_257delTATT	p.Leu85fs	Gene_gene1265	gene1265
	1,224,178	C	CT	423	c.1164dupA	p.Ala389fs	Gene_gene1275	gene1275
	1,245,304	TA	T	387	c.204delT	p.Phe68fs	BGV76_RS06470	Transcript_gene1292
	1,252,008	A	AG	351	c.377dupG	p.Val127fs	BGV76_RS06515	Transcript_gene1301
	1,408,360	CG	C	400	c.1306delC	p.Arg436fs	Gene_gene1448	gene1448
	1,472,109	CT	C	425	c.55delA	p.Ser19fs	Gene_gene1508	gene1508
	1,494,271	AT	A	581	c.355delA	p.Ile119fs	Gene_gene1538	gene1538
	1,496,486	T	TTA	673	c.220_221insAT	p.Ser74fs	Gene_gene1541	gene1541
	1,567,158	C	CG	445	c.968dupG	p.Arg324fs	Gene_gene1609	gene1609
	1,570,436	C	CG	448	c.230dupG	p.Ser78fs	Gene_gene1612	gene1612
	1,578,725	C	CG	438	c.749dupG	p.His251fs	Gene_gene1621	gene1621
	1,589,275	C	CG	457	c.108dupC	p.Asp37fs	BGV76_RS08160	Transcript_gene1630
	1,598,314	A	AG	428	c.1433dupG	p.Gly479fs	Gene_gene1636	gene1636
	1,617,562	CA	C	530	c.327delA	p.Lys109fs	BGV76_RS08260	Transcript_gene1650
	1,662,103	G	GAA	41	c.37_38insAA	p.Gly13fs	BGV76_RS08455	Transcript_gene1689
	1,662,104	GGC	G	49	c.39_40delGC	p.Gly13fs	BGV76_RS08455	Transcript_gene1689
	1,664,328	ATATTTCC	A	62	c.537_543delTATTTCC	p.Asp179fs	BGV76_RS08465	Transcript_gene1691
	1,664,342	G	GCC	75	c.550_551insCC	p.Glu184fs	BGV76_RS08465	Transcript_gene1691
	1,664,344	A	AACGAG	84	c.553_554insCGAGA	p.Lys185fs	BGV76_RS08465	Transcript_gene1691
	1,673,647	G	GC	435	c.284dupG	p.Cys95fs	Gene_gene1701	gene1701
	1,704,505	AC	A	464	c.29delG	p.Arg10fs	Gene_gene1732	gene1732
	1,729,123	CT	C	455	c.559delA	p.Arg187fs	BGV76_RS08790	Transcript_gene1756
	1,729,237	AATTGAAATGACAATTCCGGCCAGTAATCCTTTTTAACGAAAGCGTACTGGTAGAGGATGTCTAAGCCGTTATGGGTGATGGCAAG	A	497	c.361_445delCTTGCCATCACCCATAACGGCTTAGACATCCTCTACCAGTACGCTTTCGTTAAAAAGGATTACTGGCCGGAATTGTCATTTCAAT	p.Leu121fs	BGV76_RS08790	Transcript_gene1756
	1,730,794	A	AC	413	c.104dupC	p.Ter36fs	Gene_gene1758	gene1758
	1,731,126	CT	C	415	c.188delA	p.Lys63fs	BGV76_RS08805	Transcript_gene1759
	1,925,766	T	TC	385	c.1575dupG	p.Ile526fs	Gene_gene1960	gene1960
	1,957,489	G	GGC	467	c.195_196dupGC	p.His66fs	BGV76_RS09960	Transcript_gene1990
	1,957,495	CCT	C	466	c.203_204delCT	p.Ser68fs	BGV76_RS09960	Transcript_gene1990
	1,977,748	T	TG	414	c.1034dupG	p.Phe346fs	Gene_gene2007	gene2007
Stop_gained	348,140	C	T	339	c.73C>T	p.Gln25*	BGV76_RS01850	Transcript_gene368
	1,076,573	C	T	358	c.184C>T	p.Gln62*	BGV76_RS05585	Transcript_gene1115
	1,094,442	G	A	401	c.220C>T	p.Arg74*	BGV76_RS05690	Transcript_gene1136
	1,094,621	C	T	270	c.41G>A	p.Trp14*	BGV76_RS05690	Transcript_gene1136
	1,094,634	C	A	251	c.28G>T	p.Glu10*	BGV76_RS05690	Transcript_gene1136
	1,175,482	G	A	409	c.976C>T	p.Gln326*	BGV76_RS06175	Transcript_gene1233
	1,191,164	C	T	1001	c.70C>T	p.Gln24*	BGV76_RS06240	Transcript_gene1246
	1,192,075	C	T	892	c.502C>T	p.Arg168*	BGV76_RS06245	Transcript_gene1247
	1,293,932	C	T	1589	c.847C>T	p.Gln283*	BGV76_RS06730	Transcript_gene1344
	1,401,457	C	T	439	c.52C>T	p.Gln18*	BGV76_RS07210	Transcript_gene1440
	1,412,900	G	A	320	c.847C>T	p.Gln283*	BGV76_RS07270	Transcript_gene1452
	1,469,046	C	T	488	c.202C>T	p.Gln68*	BGV76_RS07540	Transcript_gene1506
	1,472,072	A	C	433	c.93T>G	p.Tyr31*	Gene_gene1508	gene1508
	1,599,733	C	A	434	c.2846C>A	p.Ser949*	Gene_gene1636	gene1636
	1,661,887	T	G	152	c.588T>G	p.Tyr196*	BGV76_RS08450	Transcript_gene1688
	1,663,750	A	T	254	c.883A>T	p.Arg295*	BGV76_RS08460	Transcript_gene1690
	1,673,835	G	A	439	c.97C>T	p.Gln33*	Gene_gene1701	gene1701
	1,705,464	G	T	411	c.4G>T	p.Glu2*	Gene_gene1734	gene1734
	1,729,568	G	A	447	c.115C>T	p.Arg39*	BGV76_RS08790	Transcript_gene1756
Frameshift_variant & stop_gained	328,669	G	GGCTCTACACTAAATCTTGTTGATGGATTACCATCGTGGTGATTCGTTGACAGGAT	294	c.596_597insATCCTGTCAACGAATCACCACGATGGTAATCCATCAACAAGATTTAGTGTAGAGC	p.His199fs	BGV76_RS01765	Transcript_gene351
	619,738	G	GATACTAAGCTT	50	c.5_6insATACTAAGCTT	p.Ser2fs	Gene_gene646	gene646
	664,789	T	TCCTAAATTGCAAGATTAAGTGAGCCACCCGGCCACGGGAG	93	c.208_209insCCTAAATTGCAAGATTAAGTGAGCCACCCGGCCACGGGAG	p.Leu70fs	BGV76_RS03485	Transcript_gene695
	1,021,085	A	AATCTACTGCTTAATCTTAGAACGTAAACTTC	57	c.1643_1644insGAAGTTTACGTTCTAAGATTAAGCAGTAGAT	p.Ter549fs	BGV76_RS05335	Transcript_gene1065
	1,175,869	C	CTGAAGTGAACCCCCGAGATTGGACAACAATCTCGGGGGTTTTATTATGA	207	c.588_589insTCATAATAAAACCCCCGAGATTGTTGTCCAATCTCGGGGGTTCACTTCA	p.Ala197fs	BGV76_RS06175	Transcript_gene1233
	1,311,946	T	TGGTTATGTCCGTATAATTGGTGTAAATTCTAAATAGGACTTTGTGAA	419	c.99_100insTTCACAAAGTCCTATTTAGAATTTACACCAATTATACGGACATAACC	p.Lys34fs	BGV76_RS06805	Transcript_gene1359
	1,412,781	G	GTGGTCAACTGAGCGTATTCTGCTGGGTATTCTTGCCCGTACTTTTC	127	c.965_966insGAAAAGTACGGGCAAGAATACCCAGCAGAATACGCTCAGTTGACCA	p.Phe322fs	BGV76_RS07270	Transcript_gene1452
	1,704,983	T	TCAAAGGGCAAGAAGTCCTGTTGCTGGTAGGTGGCCAGGGGCTG	199	c.346_347insCAGCCCCTGGCCACCTACCAGCAACAGGACTTCTTGCCCTTTG	p.Gln116fs	BGV76_RS08675	Transcript_gene1733
Stop_lost& splice_region_variant	2,72,047	A	G	16	c.470A>G	p.Ter157Trpext*?	BGV76_RS01460	Transcript_gene290
	1,192,261	T	C	935	c.688T>C	p.Ter230Glnext*?	BGV76_RS06245	Transcript_gene1247
	1,705,777	A	T	398	c.317A>T	p.Ter106Leuext*?	Gene_gene1734	gene1734
	1,948,678	A	C	385	c.399A>C	p.Ter133Tyrext*?	BGV76_RS09915	Transcript_gene1981
Frameshift_variant & stop_lost	1,476,796	CATAA	C	399	c.192_195delATAA	p.Lys64fs	Gene_gene1517	gene1517
	1,832,671	ATTTGCGGCAGCTTGTTCAAACTGGCGGTTCAAGGTATTCTGTGCTTCATAAGTTTCCTACTCCTGCACACGTTTTCTC	A	959	c.-1_74delGAGAAAACGTGTGCAGGAGTAGGAAACTTATGAAGCACAGAATACCTTGAACCGCCAGTTTGAACAAGCTGCCGCAAA	p.Lys1fs	Gene_gene1865	gene1865
Start_lost& inframe_deletion	669,110	CCTTTAAGCGCATCAGA	C	396	c.-1_12delTCTGATGCGCTTAAAG	p.Met1_Lys4del	BGV76_RS03520	Transcript_gene702
Stop_gained & disruptive_inframe_insertion	1,286,056	C	CGGTTATGTCCGTATAATTGGTGTAAATTCTAAATAGGACTTTGTGAAA	269	c.125_126insGGTTATGTCCGTATAATTGGTGTAAATTCTAAATAGGACTTTGTGAAA	p.Thr27_Ala42dup	BGV76_RS06700	Transcript_gene1338

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
