# Peer review of "Comparative Genomic Analysis and Physiological Properties of Limosilactobacillus fermentum SMFM2017-NK2 with Ability to Inflammatory Bowel Disease"

_microorganisms, 2023, doi:10.3390/microorganisms11030547_

Round 1
Reviewer 1 Report
This paper investigated the effect of lactic acid bacteria on alleviating IBD in the DDS-induced mice. The authors concluded that L. fermentum SMFM2017-NK2 effectively alleviates colitis. The biochemical characteristics, cytotoxicity, and heat stability, and genomic analysis of L. fermentum SMFM2017-NK2 suggest that it could be a novel probiotic strain. The manuscript is generally well written however, I have some concerns about the present manuscript.
Line 93 to 96. Please specific the route of administration of DDS.
Line 95. The authors stated that lactic acid bacteria were administered after colitis was induced, please state clearly if day 1 was the first day of DDS treatment or first day of lactic acid bacteria administration? The use of “after colitis was induced” was a bit confusing here.
Line 97, change “and” to “or”
Line 151. Please indicate the number of replicates in the analysis of biochemical characteristics.
Line 152. Please specific the indicative criteria of effectiveness on colitis.
Line 307. In Fig 1, please include the data on day 13. What do a, ab and b indicate?
The experimental design has 6 groups of 8 mice. When reporting data, all groups should be reported. In Fig 2 and Table 2, data from the control and normal groups were missing.
Line 400-401. The style of writing is confusing. Please revise the sentence structure.
Figure 3. What do a, ab, b and c indicate? The significance level must be indicated clearly.
Discuss the possible mechanism of DDS-induced colitis
The author combined the results and discussions sections and in general, the data was not discussed extensively.
Reviewer 2 Report
This retrospective, observational research aimed to study the anti-inflammatory effect of Latilactobacillus sakei SMFM2017-NK1 (LS1), L. sakei SMFM2017-NK3 (LS2), and Limosilactobacillus fermentum SMFM2017-NK2 (LF) on colitis model. The results showed that L. fermentum SMFM2017-NK2 effectively alleviates colitis, suggesting that it could be a probiotic strain. In addition, the comparative genomic analysis result also indicates that L. fermentum SMFM2017-NK2 is a novel strain. Therefore, L. fermentum SMFM2017-NK2 might be appropriate for anti-inflammation in the colon. After reading the manuscript carefully, I can tell you that the information it brings to readers is reasonable and interesting. The work will make a major contribution to the field of Food Microbiology. In my opinion, there are several minor revisions for this manuscript.
1. English, grammar and typos need to be further improved.
2. Please add some lastest references in the manuscript, especially references in recent two years. Furthermore, format of references needs to be unified, for example References 32, 41.
3. Although the safety evaluation, such as antibiotics resistance, presence of toxic genes, production of D-lactate, bile salt deconjugation, etc. were not examined, they should be further discussed in the Discuss section.
4. Line 419-420, Figure 3. Relative gene expressions of inflammatory cytokine [TNF-α (A), IL-1β (), IL-6 (), IFN-γ (), iNOS () and COX-2 ()] in colon. () have missed some information?
5. In this study, gene expression of proinflammatory cytokines (TNF-α, IL-1β, IL-6, and IFN-γ) and inflammation-related mediators (iNOS and COX-2) was analyzed, however, it is beeter to detect the protein expression of these genes.
6. Line 384-388, histopathological features of colon. Magnification or scale should be added.
7. In Table 2, what is the scoring standard to evaluate histomorphological score of colon?
8. In the Discussion section, the authors should combine the current research results for further discussion.
Round 2
Reviewer 1 Report
Thank you for the revised version. The changes look fine to me but I have some further concern on the statistical analysis of the data.
1. The Mann-Whitney U test is used to compare differences between two independent groups when the dependent variable is either ordinal or continuous, but not normally distributed. In the current experiment design, mice were randomly divided into 6 groups. Apart from the normal (normal mice administered with PBS) and control (colitis -induced mice) groups, mice were orally administered 2×10 106 8 CFU/day 107 of L. sakei SMFM2017 -NK1 (LS1), L. sakei SMFM2017 -NK3 (LS2), and or L. fermentum SMFM2017 -NK2 (LF), respectively, from the first day of DSS treatment . The author stated that L. rhamnosus GG (LGG) was also administered, which is known to alleviate colitis, to compare its anti-inflammatory effect with the other strains. Thus, the comparison here was not between two independent groups, but between 6 groups, the use of Mann-Whitney U test is questionable.
2. The authors responded that the difference between the normal and control groups was firstly analyzed by the weight, length of colon, and DAI score, and the authors decided that inflammation was well induced. After then, the authors analyzed histopathological features only for the lactic acid bacteria-treated groups to compare the effect on colitis among the treated groups. Why don’t the authors assign points to the normal and control group and show the representative images of the normal and control group?
